# AI Feynman 2.0: Pareto-optimal symbolic regression exploiting graph modularity

**Silviu-Marian Udrescu**[1], **Andrew Tan**[1], **Jiahai Feng**[1], **Orisvaldo Neto**[1], **Tailin Wu**[2] **& Max Tegmark**[1,3]
[1]MIT Dept. of Physics and Institute for AI & Fundamental Interactions, Cambridge, MA, USA
[2]Stanford Dept. of Computer Science, Palo Alto, CA, USA
[3]Theiss Research, La Jolla, CA, USA
[1]{sudrescu, aktan, fjiahai, oris,tegmark}@mit.edu, [2]tailin@cs.stanford.edu

## Abstract

We present an improved method for symbolic regression that seeks to fit data to formulas that are Pareto-optimal, in the sense of having the best accuracy for a given complexity. It improves on the previous state-of-the-art by typically being orders of magnitude more robust toward noise and bad data, and also by discovering many formulas that stumped previous methods. We develop a method for discovering generalized symmetries (arbitrary modularity in the computational graph of a formula) from gradient properties of a neural network fit. We use normalizing flows to generalize our symbolic regression method to probability distributions from which we only have samples, and employ statistical hypothesis testing to accelerate robust brute-force search.

## 1 Introduction

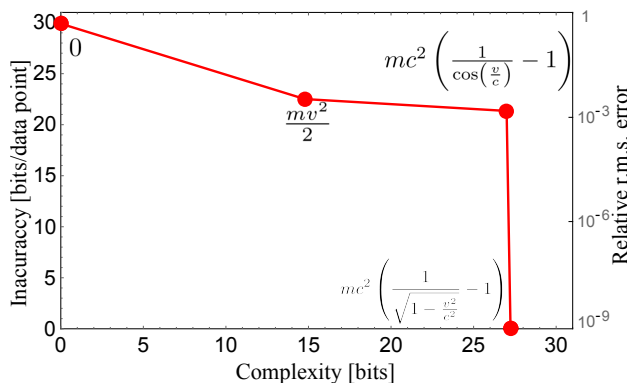

Figure 1: Our symbolic regression of data on how kinetic energy depends on mass, velocity and the speed of light discovers a Pareto-frontier of four formulas that are each the most accurate given their complexity. Convex corners reveal particularly useful formulas, in this case Einstein's formula and the classical approximation $mv^2/2$.

A central challenge in science is *symbolic regression*: discovering a symbolic expression that provides a simple yet accurate fit to a given data set. More specifically, we are given a table of numbers, whose rows are of the form $\{x_1, ..., x_n, y\}$ where $y = f(x_1, ..., x_n)$, and our task is to discover the correct symbolic expression (composing mathematical functions from a user-provided set) for the unknown mystery function $f$, optionally including the complication of noise and outliers. Science

aside, symbolic regression has the potential to replace some inscrutable black-box neural networks by simple yet accurate symbolic approximations, helping with the timely goal of making high-impact AI systems more interpretable and reliable [1–5].

Symbolic regression is difficult because of the exponentially large combinatorial space of symbolic expressions. Traditionally, it has relied on human intuition, leading to the discovery of some of the most famous formulas in science. More recently, there has been great progress toward fully automating the process [6–26], and open-source software now exists that can discover quite complex physics equations by combining neural networks with techniques inspired by physics and information theory [25]. Although [25] achieved state-of-the-art performance using a neural network approximation of the unknown function to discover simplifying function properties, it did so in an unprincipled and ad hoc way that we replace by a general, principled and much more effective method, incorporating four main contributions:

1. We recursively exploit modularity in the function's computational graph. While [25] discovered merely two types of graph modularity (symmetry and separability) involving merely four particular bivariate functions ($+$, $-$, $\times$ and $\div$), our method has the potential to discover *any* graph modularity involving *any* functions of $n = 2, 3, ...$ variables, by examining gradients of the neural network fit.

2. Instead of concluding that a candidate function or graph decomposition is good because the fitting accuracy exceeds arbitrary hyperparameter-determined thresholds, we eliminate these hyperparameters and use a Pareto-frontier (of description-length complexity versus inaccuracy) to prune our search over candidate expressions by discarding all candidates not on the frontier, improving robustness to noise and bad data.

3. Instead of simply rejecting formula candidates using $L_\infty$-norm (rejecting as soon as the error for a single data point crosses a threshold), we reject using statistical hypothesis testing, further improving robustness.

4. We use normalizing flows to enable symbolic regression of probability distributions from samples.

This enables more complex formulas to be discovered and improving noise robustness by 1-3 orders of magnitude. We describe our symbolic regression algorithm (which is publicly available[1]) in Section 2 and test it with numerical experiments in Section 3.

## 2   Method

Our symbolic regression algorithm uses a divide-and-conquer approach as in [25]. We directly solve a mystery in two base cases: if the mystery function $f(x_1, ..., x_n)$ is a low-order polynomial or if it is simple enough to be discovered by brute-force search. Otherwise, we recursively try the strategies that we will now describe for replacing it by one or more simpler mysteries, ideally with fewer input variables.

### 2.1   Leveraging graph modularity against the curse of dimensionality

When we define and evaluate a mathematical function, we typically represent it as composed of some basis set $S$ of simpler functions. As illustrated in Figure 2 (middle panel), this representation can be specified as a graph whose nodes contain elements of $S$. The most popular basis functions in the scientific literature tend to be functions of two variables (such as $+$ or $\times$), one variable (such as $\sin$ or $\log$) or no variables (constants such as $2$ or $\pi$). For many functions of scientific interest, this graph is *modular* in the sense that it can be partitioned in terms of functions with fewer input variables, as in Figure 2 (right panel).

A key strategy of our symbolic regression algorithm is to recursively discover such modularity, thereby reverse-engineering the computational graph of a mystery function, starting with no information about it other than an input-output data table. This is useful because there are exponentially many ways to combine $n$ basis functions into a module, making it extremely slow and difficult for brute-force or

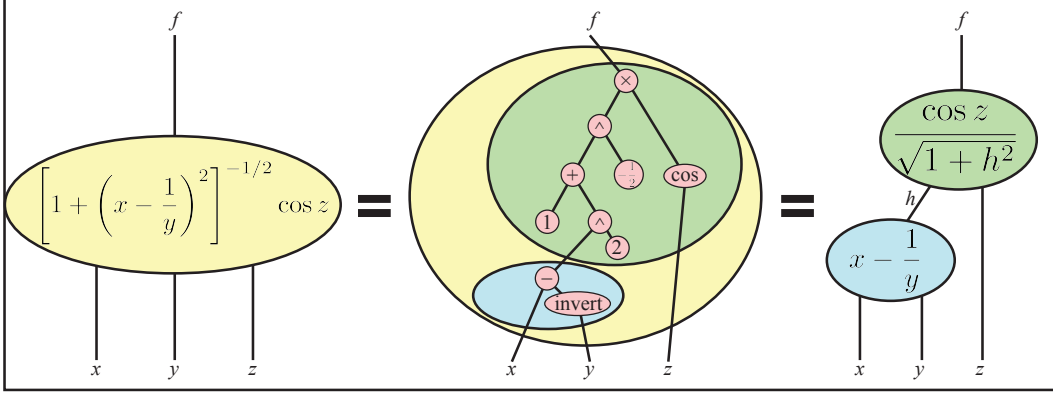

Figure 2: All functions can be represented as tree graphs whose nodes represent a set of basic functions (middle panel). Using a neural network trained to fit a mystery function (left panel), our algorithm seeks a decomposition of this function into others with fewer input variables (right panel), in this case of the form $f(x, y, z) = g[h(x, y), z]$ .

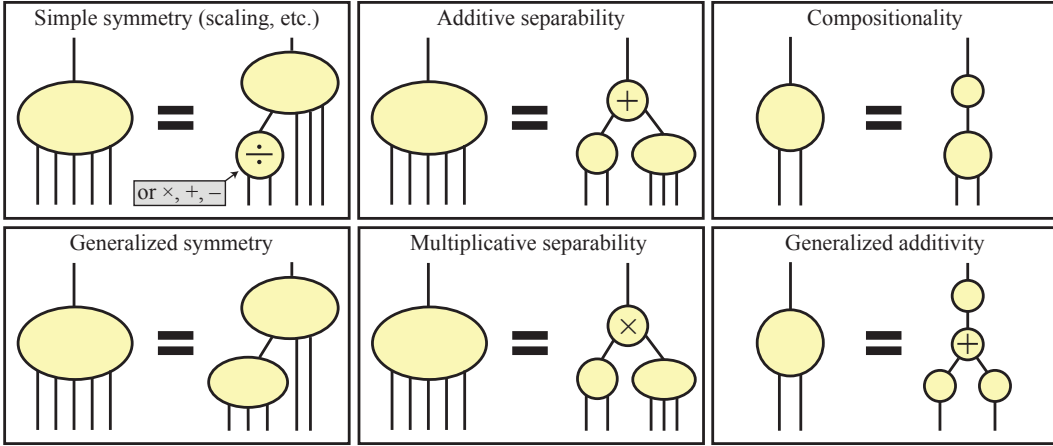

Figure 3: Examples of graph modularity that our algorithm can auto-discover. Lines denote real-valued variables and ovals denote functions, with larger ones being more complex.

genetic algorithms to discover the correct function when $n$ is large. Our divide-and-conquer approach of first breaking the function into smaller modules with smaller $n$ that can be solved separately thus greatly accelerates the solution. We implement this modularity discovery algorithm in two steps:

1. Use the user-provided data table to train a neural network $f_{\mathrm{NN}}(\mathbf{x})$ that accurately approximates the mystery function $f(\mathbf{x})$.

2. Perform numerical experiments on $f_{\mathrm{NN}}(\mathbf{x})$ to discover graph modularity.

Specifically, we test for the six types of graph modularity illustrated in Figure 3 and listed in Table 1, and choose between the discovered candidates as described in Section 2.2. Our method for discovering separability is described in [25]. As we will see below, all our other types of graph modularity (compositionality, symmetry and generalized additivity) can be revealed by $\nabla f$, the gradient of our mystery function $f$.

**Compositionality**   Let us first consider the case of *compositionality* (Figure 3, top right), where $f(\mathbf{x}) = g(h(\mathbf{x}))$ and $h$ is a scalar function simpler than $f$ in the sense of being expressible with a smaller graph as in Figure 3. By the chain rule, we have

$$\nabla f(\mathbf{x}) = g'(h(\mathbf{x}))\nabla h(\mathbf{x}), \quad \text{so} \quad \widehat{\nabla f} = \pm \widehat{\nabla h}, \tag{1}$$

Table 1: Simplification strategies

| Name | Property | Action |
|---|---|---|
| Negativity | $f(x_1, x_2, ...) < 0$ | Solve for $g \equiv -f$ |
| Positivity | $f(x_1, x_2, ...) > 0$ | Solve for $g \equiv \ln f$ |
| Additive separability | $f(x_1, ..., x_k, x_{k+1}, ..., x_n) = $ $g(x_1, ..., x_k) + h(x_{k+1}, ..., x_n)$ | Solve for $g$ & $h$ |
| Multiplicative separability | $f(x_1, ..., x_k, x_{k+1}, ..., x_n) = $ $g(x_1, ..., x_k)h(x_{k+1}, ..., x_n)$ | Solve for $g$ & $h$ |
| Simple symmetry | $f(x_1, x_2, ...) = g(x_1 \odot x_2, ...), \quad \odot \in \{+, -, \times, /\}$ | Solve for $g$ |
| Compositionality | $f(x_1, ..., x_n) = g(h(x_1, ..., x_n)), \quad h$ simpler than $f$ | Find $h$ with $\nabla h \propto \nabla f$ |
| Generalized symmetry | $f(x_1, ..., x_k, x_{k+1}, ..., x_n) = $ $g[h(x_1, ..., x_k), x_{k+1}, ..., x_n]$ | Find $h$ satisfying $\frac{\partial h}{\partial x_i} \propto \frac{\partial f}{\partial x_i}, i = 1, ..., k$ |
| Generalized additivity | $f(x_1, x_2) = F[g(x_1) + h(x_2)]$ | Solve for $F$, $g$ & $h$ |
| Zero-snap | $\tilde{f}$ has numerical parameters $\mathbf{p}$ | Replace $p_i$ by 0 |
| Integer snap | $\tilde{f}$ has numerical parameters $\mathbf{p}$ | Round $p_i$ to integer |
| Rational snap | $\tilde{f}$ has numerical parameters $\mathbf{p}$ | Round $p_i$ to fraction |
| Reoptimize | $\tilde{f}$ has numerical parameters $\mathbf{p}$ | Reoptimize $\mathbf{p}$ to minimize inaccuracy |

where hats denote unit vectors: $\widehat{\nabla f} \equiv \nabla f / |\nabla f|$, *etc.* This means that if we can discover a function $h$ whose gradient is proportional to that of $f$ (which we will describe a process for in Section 2.2), then we can simply replace the variables $\mathbf{x}$ in the original mystery data table by the single variable $h(\mathbf{x})$ and recursively apply our AI Feynman algorithm to the new one-dimensional symbolic regression problem of discovering $g(h)$.

**Generalized symmetry** Let us now turn to *generalized symmetry* (Figure 3, bottom left), where $k$ of the $n$ arguments enter only via some scalar function $h$ of them. Specifically, we say that an $f$ has generalized symmetry if the $n$ components of the vector $\mathbf{x} \in \mathbb{R}^n$ can be split into groups of $k$ and $n - k$ components (which we denote by the vectors $\mathbf{x}' \in \mathbb{R}^k$ and $\mathbf{x}'' \in \mathbb{R}^{n-k}$) such that $f(\mathbf{x}) = f(\mathbf{x}', \mathbf{x}'') = g[h(\mathbf{x}'), \mathbf{x}'']$ for some function $g$. By the chain rule, we have

$$\nabla_{\mathbf{x}'} f(\mathbf{x}', \mathbf{x}'') = g_1[h(\mathbf{x}'), \mathbf{x}''] \nabla h(\mathbf{x}'), \quad \text{so} \quad \widehat{\nabla_{\mathbf{x}'} f} = \pm \widehat{\nabla h}, \tag{2}$$

where $g_1$ denotes the derivative of $g$ with respect to its first argument. This means that $\widehat{\nabla_{\mathbf{x}'} f}(\mathbf{x}', \mathbf{x}'')$ is independent of $\mathbf{x}''$, which it would not be for a generic function $f$. $\mathbf{x}''$-independence of the normalized gradients $\widehat{\mathbf{v}}(\mathbf{x}', \mathbf{x}'') \equiv \widehat{\nabla_{\mathbf{x}'} f}(\mathbf{x}', \mathbf{x}'')$ thus provides a smoking gun signature of generalized symmetry. Whereas our compositionality discovery above requires discovering an explicit function $h$, we can discover generalized symmetry without knowing $h$, thus only performing the time-consuming task of searching for an $h$ satisfying equation (2) after determining that a solution exists. The Supplementary Material details how we numerically test for $\mathbf{x}''$-independence of $\widehat{\mathbf{v}}(\mathbf{x}', \mathbf{x}'')$.

**Generalized additivity** If $f$ is a function of two variables, then we also test for *generalized additivity* (Figure 3, bottom right), where $f(x_1, x_2) = F[g(x_1) + h(x_2)]$. If we define the function

$$s(x_1, x_2) \equiv \frac{\partial f / \partial x_1}{\partial f / \partial x_2}, \quad \text{then} \quad s(x_1, x_2) = \frac{g'(x_1)}{h'(x_2)} \tag{3}$$

if $f$ satisfies the generalized additivity property. In other words, we simply need to test if $s$ is of the multiplicatively separable form $s(x_1, x_2) = a(x_1)b(x_2)$, and we do this using a variant of the separability test described in [25]. The Supplementary Material details how we perform this separability test numerically.

Table 2: Complexity definitions

| Object | Symbol | Description length $L_d$ |
|---|---|---|
| Natural number | $n$ | $\log_2 n$ |
| Integer | $m$ | $\log_2(1 + \|m\|)$ |
| Rational number | $m/n$ | $L_d(m) + L_d(n) = \log_2[(1 + \|m\|)n]$ |
| Real number | $r$ | $\log_+\left(\frac{r}{\varepsilon}\right), \quad \log_+(x) \equiv \frac{1}{2}\log_2\left(1 + x^2\right)$ |
| Parameter vector | $\mathbf{p}$ | $\sum_i L_d(p_i)$ |
| Parametrized function | $f(\mathbf{x}; \mathbf{p})$ | $L_d(\mathbf{p}) + k\log_2 n$; $n$ basis functions appear k times |

## 2.2 Robustness through recursive Pareto-optimality

As illustrated in Figure 1, the goal of our symbolic regression of a data set is to approximate $f(\mathbf{x})$ by functions $\tilde{f}(\mathbf{x})$ that are not only accurate, but also simple, in the spirit of Occam's razor. As in [10], we seek functions that are *Pareto-optimal* in the sense of there being no other function that is both simpler and more accurate. We will adopt an information-theoretical approach and use bits of information to measure lack of both accuracy and simplicity.

For *accuracy*, we wish the vector $\boldsymbol{\varepsilon}$ of prediction errors $\varepsilon_i \equiv y_i - \tilde{f}(\mathbf{x}_i)$ to be small. We quantify this not by the mean-squared error $\langle\varepsilon_i^2\rangle$ or max-error $\max|\varepsilon_i|$ as in [10, 25], but by the MEDL, the *mean error-description-length* $\langle L_d(\varepsilon_i)\rangle$ defined in Table 2. As argued in [27] and illustrated in Figure 4, this improves robustness to outliers. We analogously quantify *complexity* by the description length $L_d$ defined as in [27], summarized in Table 2.

$L_d$ can be viewed as a crude but computationally convenient approximation of the number of bits needed to describe each object, made differentiable where possible. We choose the precision floor $\epsilon \equiv 2^{-30} \sim 10^{-9}$. For function complexity, both input variables and mathematical functions (*e.g.*, cos and +) count toward $n$ and $k$. For example, the classical kinetic energy formula has $L_d(``m \times v \times v/2'') = L_d(2) + k\log_2 n = \log_2 3 + 6\log_2 4 \approx 13.6$ bits, since the formula contains $n = 4$ basis functions ($m$, $v$, $\times$ and $/$) used $k = 6$ times.

We wish to make the symbolic regression implementation of [25] more robust; it sometimes fails to discover the correct expression because of noise in the data or inaccuracies introduced by the neural network fitting. The neural network accuracy may vary strongly with $\mathbf{x}$, becoming quite poor in domains with little training data or when the network is forced to extrapolate rather than interpolate, and we desire a regression method robust to such outliers. We expect our insistence on Pareto-optimal functions in the information plane of Figure 1 to increase robustness, both because $\langle L_d(\varepsilon_i)\rangle$ is robust (Figure 4) and because noise and systematic errors are unlikely to be predictable by a simple mathematical formula with small $L_d$. More broadly, minimization of total exact description length (which $L_d$ crudely approximates) provably avoids the overfitting problem that plagues many alternative machine-learning strategies [28–30].

**Speedup by recursive Pareto frontier composition**   When recursively symbolically regressing various modules (see Figure 2), we end up with a Pareto frontier of candidate functions for each one. If there are $n_i$ functions on the $i^{\text{th}}$ frontier, then combining them all would produce $\prod_i n_i$ candidates $\tilde{f}(\mathbf{x})$ for the original function $f(\mathbf{x})$. We speed up our algorithm by Pareto-pruning after each merge step: whenever two modules are combined (via composition or multiplication, say), the resulting $n_1 n_2$ functions are pruned by removing all functions that are Pareto-dominated by another function that is both simpler and more accurate. Pruning models on the Pareto frontier significantly reduces the number of models that need to be evaluated, since in typical scenarios, the number of Pareto-optimal points grows only logarithmically with the total number of points.

**Robust speedup of brute-force graph search with hypothesis testing**   Our recursive reduction of regression mysteries into simpler ones terminates at the base case when the mystery function has only one variable and cannot be further modularized. As in [25], we subject these (and also all multivariate modules) to two solution strategies, polynomial fitting up to some degree (4 by default) and brute-force search, and then add all candidates functions to the Pareto plane and prune as above. The brute-force search would, if run forever, try all symbolic expressions by looping

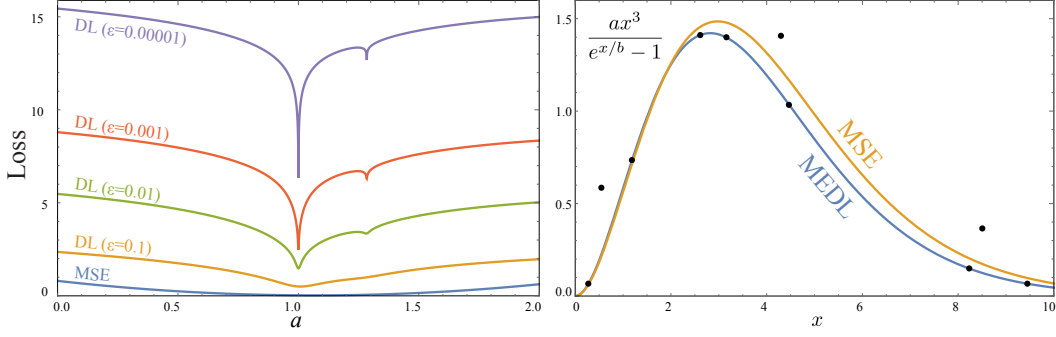

Figure 4: When fitting a function (the right panel shows the example $\frac{ax^3}{e^{x/b}-1}$) to data with outliers, minimizing mean-squared-error (MSE) biases the curve toward the outliers (here finding $a \approx 0.89$, $b \approx 1.056$), whereas minimizing mean error description length (MEDL) ignores the outliers and recovers the correct answer $a = b = 1$. Left panel compares MSE and MEDL loss functions for $b = 1$.

over ever-more-complex graphs (the middle panel of Figure 2 shows an example) and over function options for each node.

Our brute-force computation of the Pareto frontier simply tries all functions $f_k(\mathbf{x})$ ($k = 1, 2...$) in order of increasing complexity $L_d(f_k)$ and keeps only those with lower mean error-description-length $d_k \equiv \frac{1}{N} \sum_{i=1}^{N} d_{ki}$ than the previous record holder, where $d_{ki} \equiv L_d[y_i - f_k(\mathbf{x}_i)]$. When instead fitting normalized gradient vectors $\widehat{\nabla f}$ as in Section 2.1, we define $d_{ki} \equiv L_d[1 - |\hat{\mathbf{y}}_i \cdot \widehat{\nabla f_k}(\mathbf{x}_i)|]$ to handle the sign ambiguity. The bad news is that computing $d_k$ exactly is slow, requiring evaluation of $f_k(\mathbf{x}_i)$ for all $N$ data points $\mathbf{x}_i$. The good news is that this is usually unnecessary, since for the vast majority of all candidate functions, it becomes obvious that they provide a poor fit after trying merely a handful of data points. We therefore accelerate the search via the following procedure. Before starting the loop over candidate functions, we sort the data points in random order to be able to interpret the numbers $d_{ki}$ as random samples from a probability distribution whose mean is the sought-for $d_k$ and whose standard deviation is $\sigma_k$. Let $d_{k*}$ and $\sigma_{k*}$ denote the corresponding quantities that were computed for the previous best-fit function we added to the Pareto frontier. We make the simplifying approximations that $\sigma_k = \sigma_{k*}$ and that all errors are uncorrelated, so that the loss estimate from the first $m$ data points $\bar{d}_{km} \equiv \frac{1}{m} \sum_{i=1}^{m} d_{ki}$ has mean $d_k$ and standard deviation $\sigma_{k*}/\sqrt{m}$. We now test our candidate function $f_i$ on one data point at a time and reject it as soon as

$$z > \nu, \quad \text{where} \quad z \equiv \sqrt{m} \frac{\bar{d}_{km} - d_{k*}}{\sigma_{k*}}, \tag{4}$$

where $\nu$ is a hyperparameter that we can interpret as the "number of sigmas" we require to rule out a candidate function as viable when its average error exceeds the previous record holder. We find that $\nu = 10$ usually works well, generically requiring no more than a handful of evaluations $m$ per candidate function asymptotically. We can further increase robustness by increasing $\nu$ at the price of longer runtime.

**Speedup by greedy search of simplification options** We do not *a priori* know which of the modular decompositions from Figure 3 are most promising, and recursively trying all combinations of them would involve trying exponentially many options. We therefore accelerate our algorithm with a greedy strategy where at each step we compare the decomposition in a unified way and try only the most accurate one — our runtime thus grows roughly linearly with $n$, the number of input variables. $f(\mathbf{x})$ stays constant along constant-$h$ curves for generalized symmetry, simple symmetry (where $h(x, y) = x + y, x - y, xy$ or $x/y$) and generalized additivity (where $h(x, y) = a(x) + b(y)$). We thus test the accuracy of all such $h$-candidates by starting at a datapoint $\mathbf{x}_i$ and computing an error $\epsilon_i \equiv f(\tilde{\mathbf{x}}_i) - f(\mathbf{x}_i)$ for some $\tilde{\mathbf{x}}_i$ satisfying $h(\tilde{\mathbf{x}}_i) = h(\mathbf{x}_i)$. For additive and multiplicative separability, we follow [25] by examining a rectangle in parameter space and predicting $f$ at the fourth corner from the other three, defining $\epsilon_i$ as the mismatch. The supplementary material details how our test points are chosen.

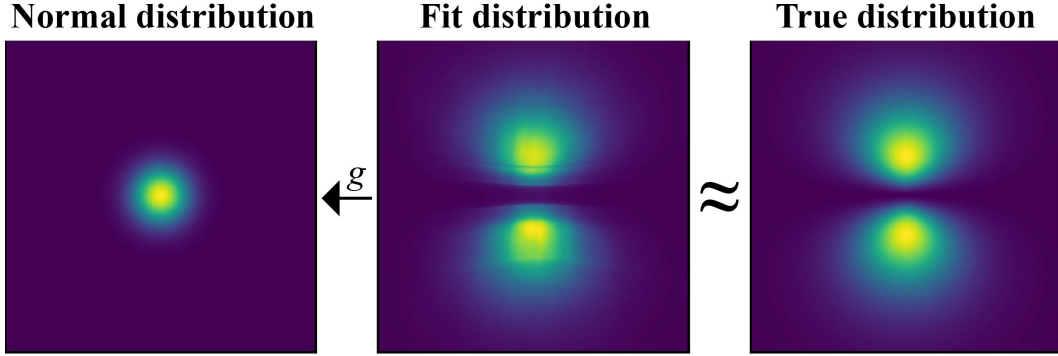

**Normal distribution**  **Fit distribution**  **True distribution**

Figure 5: A normalizing flow $g$ maps samples from a probability distribution $f$ (right) into a normal distribution (left), enabling an estimate (middle) of $f$, here illustrated for the $n = 2$, $l = 1$, $m = 0$ hydrogen orbital from Table 5.

After this greedy recursive process has terminated, we further improve the Pareto frontier in two ways. We first add models where rational numbers are replaced by reals and optimized by gradient descent to fit the data. We then add models with zero-snap, integer-snap and rational-snap from Table 1 applied to all real-valued parameters as described in [27], pruning all Pareto-dominated models after each step. For example, if there are 3 real-valued parameters, integer-snap generates 3 new models where the 1, 2 and 3 parameters closest to integers get rounded, respectively.

### 2.3 Leveraging normalizing flows to symbolic regress probability distributions

An important but more difficult symbolic regression problem is when the unknown function $f(\mathbf{x})$ is a probability distribution from which we have random samples $\mathbf{x}_i$ rather than direct evaluations $y_i = f(\mathbf{x}_i)$. We tackle this by adding preceding the regression by a step that estimates $f(\mathbf{x})$. For this step, we use the popular *normalizing flow* technique [31–35], training an invertible neural network mapping $\mathbf{x} \mapsto \mathbf{x}' \equiv g(\mathbf{x})$ such that $\mathbf{x}'$ has a multivariate normal distribution $n(\mathbf{x}')$ as illustrated in Figure 5. We then obtain our estimator $f_{\mathrm{NN}}(\mathbf{x}) = n[g(\mathbf{x})]|J|$, where $J$ is the Jacobian of $g$.

We find rational-quadratic neural spline flows (RQ-NSF) suitable for relatively low-dimensional applications due to their enhanced expressivity. Specifically, we used three steps of the RQ-NSF with RQ-NSF (C) coupling layers as described in [34], parametrized by three 16-neuron softplus layers, trained for $50,000$ epochs with the Adam optimizer. The learning rate was initialized to $3 \times 10^{-4}$ and halved every time the test loss failed to improve for 2500 epochs.

### 2.4 Neural Network training

Our neural network approximation $f_{\mathrm{NN}}$ of the mystery function $f$ is fully-connected, feed-forward neural network with 4 hidden layers of 128, 128, 64 and 64 neurons, respectively, all with *tanh* activation function. We used $80\%$ of the available data the training and the rest for validation. We used the $r.m.s.$ error loss function and the Adam optimizer with $\beta$-parameters of 0.9 and 0.999. The learning rate was initialized to 0.01 and reduced by a factor of 10 whenever the validation loss failed to improve for more than 20 epoch, until it reached $10^{-5}$. As mentioned, all our code is available at https://ai-feynman.readthedocs.io and by typing "pip install aifeynman".

## 3 Results

We now turn to quantifying the performance of our method with numerical experiments, comparing it with that of [25] which recently exceeded the previous state of-the-art performance of [10]. To quantify robustness to noise, we add Gaussian noise of standard deviation $10^r$ to $y_i$ and determine the largest integer $r < 0$ for which the method successfully discovers the correct mystery function $f(\mathbf{x})$. As seen in Table 3, our method solves 73 of 100 baseline problems from the Feynman Symbolic Regression Database [25] with $r = -1$, and is typically 1-3 orders of magnitude more robust than that of [25]. Crudely speaking, we found that adding progressively more noise shifted the most accurate

Table 3: Robustness to noise

| New robustness | | $10^{-5}$ | $10^{-4}$ | $10^{-3}$ | $10^{-2}$ | $10^{-1}$ | Total |
|---|---|---|---|---|---|---|---|
| Old robustness | $10^{-5}$ | 0 | 1 | 2 | 2 | 0 | 5 |
| | $10^{-4}$ | 0 | 1 | 3 | 5 | 12 | 21 |
| | $10^{-3}$ | 0 | 0 | 5 | 6 | 24 | 35 |
| | $10^{-2}$ | 0 | 0 | 0 | 2 | 37 | 39 |
| | Total | 0 | 2 | 10 | 15 | 73 | 100 |

Table 4: Test equations exhibiting translational symmetry $h = x \pm y$ (T), scaling symmetry $h = x/y$ (S), product symmetry $h = xy$ (P), generalized symmetry (G), multiplicative separability (M), compositionality (C) and generalized additivity (A).

| | Equation | Symmetries |
|---|---|---|
| 1 | $\delta = -5.41 + 4.9 \frac{\alpha - \beta + \gamma/\chi}{3\chi}$ | TC |
| 2 | $\chi = 0.23 + 14.2 \frac{\alpha + \beta}{3\gamma}$ | TS |
| 3 | $\beta = 213.80940889 \left(1 - e^{-0.54723748542\alpha}\right)$ | |
| 4 | $\delta = 6.87 + 11\sqrt{\alpha\beta\gamma}$ | P |
| 5 | $V = \left[R_1^{-1} + R_2^{-1} + R_3^{-1} + R_4^{-1}\right]^{-1} I_0 \cos \omega t$   (Parallel resistors) | PGSM |
| 6 | $I_0 = \frac{V_0}{\sqrt{R^2 + \left(\omega L - \frac{1}{\omega C}\right)^2}}$   (RLC circuit ) | MG |
| 7 | $I = \frac{V_0 \cos \omega t}{\sqrt{R^2 + \left(\omega L - \frac{1}{\omega C}\right)^2}}$   (RLC circuit) | MG |
| 8 | $V_2 = \left(\frac{R_2}{R_1 + R_2} - \frac{R_x}{R_x + R_3}\right) V_1$   (Wheatstone bridge) | SGMA |
| 9 | $v = c \frac{(v_1 + v_2 + v_3)/c + v_1 v_2 v_3/c^3}{1 + (v_1 v_2 + v_1 v_3 + v_2 v_3)/c^2}$   (Velocity addition) | AG |
| 10 | $v = c \frac{(v_1 + v_2 + v_3 + v_4)/c + (v_2 v_3 v_4 + v_1 v_3 v_4 + v_1 v_2 v_4 + v_1 v_2 v_3)/c^3}{1 + (v_1 v_2 + v_1 v_3 + v_1 v_4 + v_2 v_3 + v_2 v_4 + v_3 v_4)/c^2 + v_1 v_2 v_3 v_4/c^4}$   (Velocity addition) | GA |
| 11 | $z = (x^4 + y^4)^{1/4}$   ($L_4$-norm) | AC |
| 12 | $w = xyz - z\sqrt{1 - x^2}\sqrt{1 - y^2} - y\sqrt{1 - x^2}\sqrt{1 - z^2} - x\sqrt{1 - y^2}\sqrt{1 - z^2}$ | GA |
| 13 | $z = \frac{xy + \sqrt{1 - x^2 - y^2 + x^2 y^2}}{y\sqrt{1 - x^2} - x\sqrt{1 - y^2}}$ | A |
| 14 | $z = y\sqrt{1 - x^2} + x\sqrt{1 - y^2}$ | A |
| 15 | $z = xy - \sqrt{1 - x^2}\sqrt{1 - y^2}$ | A |
| 16 | $r = \frac{a}{\cot(\alpha/2) + \cot(\beta/2)}$   (Incircle) | GMAC |

formula straight upward in the Pareto plane (Figure 1) until it no longer provided any accuracy gains compared with simpler approximations.

To quantify the ability of our method to discover more complex equations, we reran it on all 17 mysteries that [25] tackled and failed to solve. We also tested a dozen new mysteries exhibiting various forms of graph modularity (see Table 4) that were all chosen before any of them were tested. Allowing at most two hours of runtime, the method of [25] solved 5 of the equations, whereas our new method solved them all, as well as four of the outstanding mysteries from [25] (rows 1-4). For these first four, our method got the numerical parameters in the right ballpark with rational approximations, then discovered their exact values through gradient descent.

To quantify the ability of our method to discover probability distributions, we tested it on samples from the ten distributions in Table 5. As seen in the table, 80% were solved, requiring between $10^2$ and $10^5$ samples $\mathbf{x}_i$. The flows trained in about 20 minutes on one CPU, scaling roughly linearly with sample size and number of network weights.

We discuss common failure modes below. Interestingly, these do not include overfitting, for multiple reasons: (1) We early-stop training when the validation loss starts increasing. (2) We avoid using our neural network (to guess symbolic functions) outside the domain where it was trained. (3) Overfitting noise would generally *reduce* apparent graph modularity, thus causing failure to discover formulas rather than discovery of spurious "overfit" formulas. (4) A key desirable feature of the minimum-description-length formalism (the information-theoretical inspiration for our method) is that it provably avoids overfitting as shown in [28, 30].

Table 5: Probability distributions and number of samples $N$ required to discover them

| Distribution Name | Probability distribution | $N$ |
|---|---|---|
| Laplace distribution | $\frac{1}{2}e^{-|x|}$ | $10^2$ |
| Beta distribution ($\alpha = 0.5$, $\beta = 0.5$) | $\frac{1}{\pi}\frac{1}{\sqrt{x(1-x)}}$ | $10^4$ |
| Beta distribution ($\alpha = 5$, $\beta = 2$) | $30x^4(1-x)$ | $10^4$ |
| Harmonic oscillator ($n = 2$, $\frac{m\omega}{\hbar} = 1$) | $\frac{2}{\sqrt{\pi}}x^2 e^{-x^2}$ | $10^5$ |
| Sinc diffraction pattern | $\frac{1}{\pi}\left(\frac{\sin x}{x}\right)^2$ | $10^4$ |
| 2D normal distribution (correlated) | $\frac{1}{\sqrt{3}\pi}e^{-\frac{2}{3}(x^2-xy+y^2)}$ | $10^3$ |
| 2D harmonic oscillator ($n = 2$, $m = 1$, $\frac{m\omega}{\hbar} = 1$) | $\frac{2}{\pi}x^2 e^{-x^2-y^2}$ | $10^5$ |
| Hydrogen orbital ($n = 1$, $l = 0$, $m = 0$) | $\frac{1}{\pi}e^{-2r}$ | $10^3$ |
| Hydrogen orbital ($n = 2$, $l = 1$, $m = 0$) | $\frac{1}{16}r^2 e^{-r}\cos^2\theta$ | - |
| Hydrogen orbital ($n = 3$, $l = 1$, $m = 0$) | $\frac{1}{729}r^2\left(4-\frac{2r}{3}\right)^2 e^{-\frac{2r}{3}}\cos^2\theta$ | - |

## 4  Conclusions

We have presented a symbolic regression method that exploits neural networks, graph modularity, hypothesis testing and normalizing flows, and released it at `https://ai-feynman.readthedocs.io`. It improves state-of-the-art performance both by being more robust towards noise and by solving harder problems, including symbolic density estimation.

Despite these advances, numerous equations remained unsolved, motivating further work. Here are some interesting failure modes we identified. Some cases failed because the form of the equation precluded our method from breaking it into small enough pieces. For example, for the equation

$$\alpha^3 e^{-\alpha}\cos\alpha\sin\alpha\left[\cos\alpha(\sin\alpha)^2 - 1\right](\beta - 5),$$

our algorithm discovered the multiplicative separability into terms including only $\alpha$ and only $\beta$. However, the remaining $\alpha$-term was too complicated to be solved in a reasonable amount of time by the brute force code, and none of the graph modularity methods apply because they only help for functions of more than one variable. For other equations, our method fails but not irreparably. For example, for the function

$$22 - 4.2\left[\cos\alpha - \tan\beta\right]\tanh\gamma/\sin\chi,$$

our code is able to discover that $\gamma$ and $\chi$ can be separated from the rest of the equation. However, given that we allow the brute force code to run for only a minute for each iteration, the expression $\tanh\gamma$ is not discovered, mainly because we did not include $\tanh$ as one of the functions used, so the brute force would have to write that as $\frac{e^{2x}-1}{e^{2x}+1}$. By allowing the code to run for longer or using with other basis functions (such as $\tanh$), the code could solve this and several other mysteries that we reported as failures. Success and failure modes are further discussed in the Supplementary Material.

There are many obvious ways in which the core ideas of this paper can be extended to further improve symbolic regression. For example, gradients can reveal more types of graph modularity than the Figure 3 examples that we exploited (e.g. where modules output more than one variable), additional simplification strategies can be included in the Pareto-optimal recursion, and flow-based regression can be used for regularized density estimation from sparse high-dimensional data. Larger and more challenging collections of science-based equations are needed to benchmark and inspire improved algorithms. A higher-level direction for improvement is to generalize the problem itself: Whereas symbolic regression takes the regression variables as given, discovering novel formulas from real-world data typically requires also the pre-regression step of mapping high-dimensional data into a low-dimensional latent space whose coordinates are promising candidates for symbolic regression; [36] provides a literature review and early steps in this direction.

Pareto-optimal symbolic regression has the power to not only discover exact formulas, but also approximate ones that are useful for being both accurate and simple. The mainstream view is that all known science formulas are such approximations. We live in a golden age of research with ever-larger datasets produced by both experiments and numerical computations, and we look forward to a future when symbolic regression is as ubiquitous as linear regression is today, helping us better understand the relations hidden in these datasets.

## Acknowledgments and Disclosure of Funding

The authors with to thank Philip Tegmark for helpful comments, and the Center for Brains, Minds, and Machines (CBMM) for hospitality.

**Funding:** This work was supported by The Casey and Family Foundation, the Ethics and Governance of AI Fund, the Foundational Questions Institute, the Rothberg Family Fund for Cognitive Science and the Templeton World Charity Foundation, Inc.

**Competing interests:** The authors declare that they have no competing interests.

**Data and materials availability:** Data and code have been publicly released at `https://ai-feynman.readthedocs.io`.

## Broader Impact

### Who may benefit from this research

Our research presumably has quite broad impact, since discovery of mathematical patterns in data is a central problem across the natural and social sciences. Given the ubiquity of *linear* regression in research, one might expect that there will significant benefits to a broad range of researchers also from more general symbolic regression once freely available algorithms get sufficiently good.

### Who may be put at disadvantage from this research

Although it is possible that some numerical modelers could get their jobs automated away by symbolic regression, we suspect that the main effect of our method, and future tools building on it, will instead be that these people will simply discover better models than today.

### Risk of bias, failure and other negative outcomes

Pareto-optimal symbolic regression can be viewed as an extreme form of lossy data compression that uncovers the simplest possible model for any given accuracy. To the extent that overfitting can exacerbate bias, such model compression is expected to help. Moreover, since our method produces closed-form mathematical formulas that have excellent interpretability compared to black-box neural networks, they make it easier for humans to interpret the computation and pass judgement on whether it embodies unacceptable bias. This interpretability also reduces failure risk.

Another risk is automation bias, whereby people overly trust a formula from symbolic regression when they extrapolate it into an untested domain. This could be exacerbated if symbolic regression promotes scientific laziness and enfeeblement, where researchers fit phenomenological models instead of doing the work of building models based on first principles. Symbolic regression should inform but not replace traditional scientific discovery.

Although the choice of basis functions biases the discoverable function class, our method is agnostic to basis functions as long as they are mostly differentiable.

The greatest potential risk associated with this work does not stem from it failing but from it succeeding: accelerated progress in symbolic regression, modularity discovery and its parent discipline, program synthesis, could hasten the arrival of artificial general intelligence, which some authors have argued humanity still lacks the tools to manage safely [5]. On the other hand, our work may help accelerate research on intelligible intelligence more broadly, and powerful future artificial intelligence is probably safer if we understand aspects of how it works than if it is an inscrutable black box.

## Footnotes

[1]Our code is can be installed by typing *pip install aifeynman* and is also available at `https://ai-feynman.readthedocs.io`.

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
