[Supplementary Material]

# AI Feynman 2.0: Pareto-optimal symbolic regression exploiting graph modularity — supplementary material

**Silviu-Marian Udrescu, Andrew Tan, Jiahai Feng, Orisvaldo Neto, Tailin Wu & Max Tegmark**
MIT Dept. of Physics & Center for Brains, Minds & Machines
Cambridge, MA 02139
sudrescu@mit.edu

Below we provide additional technical details about how we implement our method and numerical experiments.

## A  Testing for generalized symmetry

We showed that generalized symmetry can be revealed by $\widehat{\mathbf{v}}(\mathbf{x}', \mathbf{x}'')$ being independent of $\mathbf{x}''$. We will now describe how we test for such $\mathbf{x}''$-independence numerically. Given a point $\mathbf{x}_i \in \mathbb{R}^k$ from our data set, we compute a set of normalized gradients $\widehat{\mathbf{v}}_j \equiv \widehat{\mathbf{v}}_i(\mathbf{x}_i', \mathbf{x}_j'')$, where $\mathbf{x}_j'' \in \mathbb{R}^{n-k}$ correspond to a sample of $m$ other data points, and quantify the variation between then by the quantity

$$V(\mathbf{x}) \equiv 1 - \max_{|\mu|=1} \frac{1}{m} \sum_{j=1}^{m} (\widehat{\mu} \cdot \widehat{\mathbf{v}}_j)^2 = 1 - \max_{|\mu|=1} \widehat{\mu}^t \mathbf{V} \widehat{\mu}, \quad \text{where} \quad \mathbf{V} \equiv \frac{1}{m} \sum_{j=1}^{m} \widehat{\mathbf{v}}_j \widehat{\mathbf{v}}_j^t. \quad (1)$$

We can intuitively interpret the optimal $\widehat{\mu}$ as maximally aligned with the vectors $\widehat{\mathbf{v}}_j$ up to a sign. Equation (1) implies that our variation measure $V$ is simply one minus the smallest eigenvalue of $\mathbf{V}$, so $V$ ranges from 0 when all $\widehat{\mathbf{v}}_j$ are identical to $1 - \frac{1}{m}$ when all eigenvalues are equal (equal to $1/m$, since $\text{tr} \, \mathbf{V} = 1$). As illustrated in Figure 1, we compute $V(\mathbf{x}_i)$ for each subset of up to $n_g$ input variables, and select the subset with the smallest median $V(\mathbf{x}_i)$ as the most promising generalized symmetry candidate. In our numerical experiments, we set the hyperparameter $n_g = 3$ to save time, since we do not wish to consider all $2^n$ subsets for large $n$.

Figure 1: Distribution of $V(\mathbf{x}_i)$ for the function from Figure 2, revealing that evidence for the generalized symmetry $f(x, y, z) = g[h(x, y), z]$ (shaded distribution) is stronger than for $f(x, y, z) = g[h(x, z), y]$ (blue curve) or $f(x, y, z) = g[h(y, z), x]$ (red curve). The curves are shown slightly smoothed for clarity.

|                             | Schmidt & Lipson 2009 | Udrescu & Tegmark 2020 | This paper |
|-----------------------------|:---------------------:|:----------------------:|:----------:|
| FDSR basic 100              | 71%                   | **100%**               | **100%**   |
| FDSR harder 20              | 15%                   | 85%                    | **90%**    |
| 12 modular equations        | 42%                   | 42%                    | **100%**   |
| 10 probability distributions| 60%                   | 70%                    | **80%**    |

Table 6: Fraction of symbolic regression problems solved for the benchmarks in the Feynman Database for Symbolic Regression (FDRS) and this paper.

## B   Testing for generalized additivity

We showed that generalized additivity holds when the function $s(x_1, x_2)$ from Equation (3) from the main part of the paper is multiplicatively separable. We will now describe how we test for such separability numerically. $s(x_1, x_2)$ being multiplicative separable is equivalent to $f(x_1, x_2) \equiv \ln s(x_1, x_2)$ being additively separable. We numerically test the function $\ln s_{NN}(x_1, x_2)$ for additive separability using the normalized score $S$ defining

$$S[f] = \frac{|f_{,xy}|^2}{|f_{,xx}f_{,yy}| + |f_{,xy}|^2}. \tag{2}$$

It is easy to see that $S[f] = 0$ if $f$ is additively separable, and $S[f] > 0$ otherwise. If the median value of $S$ over all points $\mathbf{x}_i$ in the dataset is below a threshold $S_*$, we take this as evidence for generalized additivity and proceed as below. We found empirically that the threshold choise $S_* = 0.1$ produced robust results. It is important to use smooth (not, *e.g.*, ReLU) activation functions for this derivative-based test to be useful.

If this property holds, then we recursively apply our algorithm to the two new 1-dimensional symbolic regression problems of discovering $a(x_1)$ and $b(x_2)$. If this succeeds and we are able to discover the functions $g(x_1)$ and $h(x_2)$ by symbolically integrating our solutions $g' = a$ and $h' = 1/b$, then we have reduced the original problem to the same state as when we found compositionality above, now with $h(x_1, x_2) = g(x_1) + h(x_2)$. Just as in that case, we simply replace the variables $\mathbf{x}$ in the original mystery data table by the single variable $h(\mathbf{x})$ and recursively apply our AI Feynman algorithm to the new 1-dimensional symbolic regression problem of discovering how $f$ depends on $h$.

If we have determined that generalized additivity holds but the aforementioned method for discovering $g(x_1) + h(x_2)$ fails, we make a second attempt by training a neural network of the modular form $f_{\mathrm{NN}}(x_1, x_2) = F[g(x_1) + h(x_2)]$ to fit the data. If this succeeds, we then recursively apply our AI Feynman algorithm to the three new 1-dimensional symbolic regression problems of discovering $F$, $g$ and $h$.

## C   Further details on success and failure modes

Our paper reports which symbolic regression problems our method succeeded and failed on, as detailed in Tables 6-8. Here we add specifics on how these successes and failures occurred.

**Success definition**   Given a data set $\{x_1, ..., x_n, y\}$, we use 90% of the data to compute a Pareto-optimal set of candidate functions $\tilde{f}_i(\mathbf{x})$, then rank them based on their MEDL accuracy on the held-back 10% of the data. We count our method as successful only if the top-ranked function matches the true $f(\mathbf{x})$ exactly, or, if the definition of $f$ involves irrational numerical parameters, if these parameters are recovered to better than $0.01\%$ relative accuracy.

We considered an equation solved even if the top solution was not in the exact form presented in our tables, but mathematically equivalent. For example, our method predicted that Equation (12) in Table 4 was $w = \cos[\arccos(x) + \arccos(y) + \arccos(z)]$, which is mathematically equivalent within the domain of our provided data set, where $x, y, z \in [-1, 1]$.

For the problem of density estimation from samples, our goal was to obtain the correct normalized probability distributions. The candidate functions on the Pareto-frontier were therefore discarded unless they were non-negative and normalizable. The surviving candidates were then normalized to integrate to unity by symbolic/numerical integration to obtain the appropriate normalization constant,

and quality-ranked by the surprisal loss function

$$L_i = -\sum \log \tilde{f}_i(\mathbf{x}_k)$$

evaluated on the held-back test data.

**Success examples** Tables 7 and 8 below show the highest noise level allowing each of the 100 equations from the Feynman Database for Symbolic regression to be solved in the original paper analyzing it and in the present paper.

For many of the solved equations, the modularity discovery had to be used multiple times in order for the correct equation to be discovered, reflecting the power of the recursive algorithm. For example, for the quadruple velocity addition equation in Table 4, generalized symmetry was exploited twice. First, the code discovered that the first two velocities only enter in the combination $\frac{v_1+v_2}{1+v_1v_2}$, and these two variables were replaced by a new variable $v_{12}$. The same method then discovered that $v_{12}$ and $v_3$ only enter in that same combination $\frac{v_{12}+v_3}{1+v_{12}v_3}$, and thus the initial 3 variables $v_1$, $v_2$ and $v_3$ were replaced by a single variable $v_{123}$. Now the remaining equation had only 2 variables left, and was solved by brute force. In principle, this recursive method can be used to discover relativistic addition of an arbitrary number of velocities, by reducing the number of variables by one at each step.

**Failure examples** Some of the most obvious failure modes we discussed in the conclusions of the main text. Here we discuss some more subtle failure modes. Firstly, it is worth noting that our definition of complexity is dependent on the chosen set of operations and does not always match our intuition. For example, in fitting the probability distribution

$$p(r,\theta) = \frac{1}{16}r^2 e^{-r} \cos^2\theta$$

of electron positions in the $n = 2$, $l = 1$, $m = 0$ hydrogen orbital, solutions with $\theta$-dependence $\cos(\cos\theta)$ are preferred over $\cos^2\theta$. This is because, up to additive and multiplcative prefactors, the two formulas differ by at most approximately $2 \times 10^{-2}$ over our parameter range, but given a set of operations that includes only $\{\times, \cos\}$ denoted by "$*$" and "$C$" respectively in reverse Polish notation, $\cos(\cos\theta))$ (encoded as "$xCC$") is simpler than $\cos^2\theta$ (encoded as "$xCxC*$"). In the presence of the imprecisions introduced by the normalizing flow, we were unable to perform the density estimation a level at which the accuracy for the correct $\cos^2\theta$ was preferred over the simpler alternative.

Furthermore, more interpretable approximations (e.g. Taylor expansions) are not always favored by our definition of complexity. For example, in Figure 1, the unfamiliar solution

$$mc^2\left(\frac{1}{\cos v/c} - 1\right)$$

intermediate to the more familiar $\frac{mv^2}{2}$ and $mc^2\left(\frac{1}{\sqrt{1-v^2/c^2}} - 1\right)$ of can be understood as a fourth-order expansion about $v = 0$ of the exact formula. Specifically, $mc^2\left(\frac{1}{\sqrt{1-v^2/c^2}} - 1\right) = \frac{mv^2}{2} + \frac{3mv^4}{8c^2} + O(v^6)$, and $mc^2\left(\frac{1}{\cos v/c} - 1\right) = \frac{mv^2}{2} + \frac{5mv^4}{24c^2} + O(v^6)$. The Taylor expansions themselves are not preferred for reasons of complexity.

| Feynman eq. | Equation | Old Noise tolerance | New Noise tolerance |
|---|---|---|---|
| I.6.20a | $f = e^{-\theta^2/2}/\sqrt{2\pi}$ | $10^{-2}$ | $10^{-1}$ |
| I.6.20 | $f = e^{-\frac{\theta^2}{2\sigma^2}}/\sqrt{2\pi\sigma^2}$ | $10^{-4}$ | $10^{-2}$ |
| I.6.20b | $f = e^{-\frac{(\theta-\theta_1)^2}{2\sigma^2}}/\sqrt{2\pi\sigma^2}$ | $10^{-4}$ | $10^{-2}$ |
| I.8.14 | $d = \sqrt{(x_2-x_1)^2 + (y_2-y_1)^2}$ | $10^{-4}$ | $10^{-1}$ |
| I.9.18 | $F = \frac{Gm_1m_2}{(x_2-x_1)^2+(y_2-y_1)^2+(z_2-z_1)^2}$ | $10^{-5}$ | $10^{-3}$ |
| I.10.7 | $m = \frac{m_0}{\sqrt{1-\frac{v^2}{c^2}}}$ | $10^{-4}$ | $10^{-2}$ |
| I.11.19 | $A = x_1y_1 + x_2y_2 + x_3y_3$ | $10^{-3}$ | $10^{-1}$ |
| I.12.1 | $F = \mu N_n$ | $10^{-3}$ | $10^{-1}$ |
| I.12.1a | $K = \frac{1}{2}m(v^2 + u^2 + w^2)$ | $10^{-4}$ | $10^{-1}$ |
| I.12.2 | $F = \frac{q_1q_2}{4\pi\epsilon r^2}$ | $10^{-2}$ | $10^{-1}$ |
| I.12.4 | $U = \frac{q_1}{4\pi\epsilon r^2}$ | $10^{-2}$ | $10^{-1}$ |
| I.12.5 | $F = q_2 E_f$ | $10^{-2}$ | $10^{-1}$ |
| I.12.11 | $F = q(E_f + Bv\sin\theta)$ | $10^{-3}$ | $10^{-1}$ |
| I.13.12 | $U = Gm_1m_2(\frac{1}{r_2} - \frac{1}{r_1})$ | $10^{-4}$ | $10^{-1}$ |
| I.14.3 | $U = mgz$ | $10^{-2}$ | $10^{-1}$ |
| I.14.4 | $U = \frac{k_{spring}x^2}{2}$ | $10^{-2}$ | $10^{-1}$ |
| I.15.3x | $x_1 = \frac{x-ut}{\sqrt{1-u^2/c^2}}$ | $10^{-3}$ | $10^{-3}$ |
| I.15.3t | $t_1 = \frac{t-ux/c^2}{\sqrt{1-u^2/c^2}}$ | $10^{-4}$ | $10^{-3}$ |
| I.15.1 | $p = \frac{m_0 v}{\sqrt{1-v^2/c^2}}$ | $10^{-4}$ | $10^{-1}$ |
| I.16.6 | $v_1 = \frac{u+v}{1+uv/c^2}$ | $10^{-3}$ | $10^{-2}$ |
| I.18.4 | $r = \frac{m_1 r_1 + m_2 r_2}{m_1+m_2}$ | $10^{-2}$ | $10^{-1}$ |
| I.18.12 | $\tau = rF\sin\theta$ | $10^{-3}$ | $10^{-1}$ |
| I.18.14 | $L = mrv\sin\theta$ | $10^{-3}$ | $10^{-1}$ |
| I.24.6 | $E = \frac{1}{4}m(\omega^2 + \omega_0^2)x^2$ | $10^{-4}$ | $10^{-1}$ |
| I.25.13 | $V_e = \frac{q}{C}$ | $10^{-2}$ | $10^{-1}$ |
| I.26.2 | $\theta_1 = \arcsin(n\sin\theta_2)$ | $10^{-2}$ | $10^{-1}$ |
| I.27.6 | $f_f = \frac{1}{\frac{1}{d_1}+\frac{n}{d_2}}$ | $10^{-2}$ | $10^{-1}$ |
| I.29.4 | $k = \frac{\omega}{c}$ | $10^{-2}$ | $10^{-1}$ |
| I.29.16 | $x = \sqrt{x_1^2 + x_2^2 - 2x_1x_2\cos(\theta_1-\theta_2)}$ | $10^{-4}$ | $10^{-3}$ |
| I.30.3 | $I_* = I_{*0}\frac{\sin(n\theta/2)}{\sin(\theta/2)}$ | $10^{-3}$ | $10^{-3}$ |
| I.30.5 | $\theta = \arcsin(\frac{\lambda}{nd})$ | $10^{-3}$ | $10^{-1}$ |
| I.32.5 | $P = \frac{q^2 a^2}{6\pi\epsilon_c^3}$ | $10^{-2}$ | $10^{-1}$ |
| I.32.17 | $P = (\frac{1}{2}\epsilon c E_f^2)(8\pi r^2/3)(\omega^4/(\omega^2-\omega_0^2)^2)$ | $10^{-4}$ | $10^{-3}$ |
| I.34.8 | $\omega = \frac{qvB}{p}$ | $10^{-2}$ | $10^{-1}$ |
| I.34.10 | $\omega = \frac{1+v/c}{1-v/c}\omega_0$ | $10^{-3}$ | $10^{-2}$ |
| I.34.14 | $\omega = \frac{1+v/c}{\sqrt{1-v^2/c^2}}\omega_0$ | $10^{-3}$ | $10^{-3}$ |
| I.34.27 | $E = \hbar\omega$ | $10^{-2}$ | $10^{-1}$ |
| I.37.4 | $I_* = I_1 + I_2 + 2\sqrt{I_1 I_2}\cos\delta$ | $10^{-3}$ | $10^{-2}$ |
| I.38.12 | $r = \frac{4\pi\epsilon\hbar^2}{mq^2}$ | $10^{-2}$ | $10^{-1}$ |
| I.39.10 | $E = \frac{3}{2}p_F V$ | $10^{-2}$ | $10^{-1}$ |
| I.39.11 | $E = \frac{1}{\gamma-1}p_F V$ | $10^{-3}$ | $10^{-1}$ |
| I.39.22 | $P_F = \frac{nk_b T}{V}$ | $10^{-4}$ | $10^{-1}$ |
| I.40.1 | $n = n_0 e^{-\frac{mgx}{k_b T}}$ | $10^{-2}$ | $10^{-1}$ |
| I.41.16 | $L_{rad} = \frac{\hbar\omega^3}{\pi^2 c^2(e^{\frac{\hbar\omega}{k_b T}}-1)}$ | $10^{-5}$ | $10^{-4}$ |
| I.43.16 | $v = \frac{\mu_{drift}qV_e}{d}$ | $10^{-2}$ | $10^{-1}$ |
| I.43.31 | $D = \mu_e k_b T$ | $10^{-2}$ | $10^{-1}$ |
| I.43.43 | $\kappa = \frac{1}{\gamma-1}\frac{k_b v}{A}$ | $10^{-3}$ | $10^{-1}$ |
| I.44.4 | $E = nk_b T\ln(\frac{V_2}{V_1})$ | $10^{-3}$ | $10^{-1}$ |
| I.47.23 | $c = \sqrt{\frac{\gamma pr}{\rho}}$ | $10^{-2}$ | $10^{-1}$ |
| I.48.2 | $E = \frac{mc^2}{\sqrt{1-v^2/c^2}}$ | $10^{-5}$ | $10^{-3}$ |
| I.50.26 | $x = x_1[\cos(\omega t) + \alpha\cos(\omega t)^2]$ | $10^{-2}$ | $10^{-1}$ |

Table 7: Tested Equations, part 1

| Feynman eq. | Equation | Old Noise tolerance | New Noise tolerance |
|---|---|---|---|
| II.2.42 | $P = \frac{\kappa(T_2 - T_1)A}{d}$ | $10^{-3}$ | $10^{-1}$ |
| II.3.24 | $F_E = \frac{P}{4\pi r^2}$ | $10^{-2}$ | $10^{-1}$ |
| II.4.23 | $V_e = \frac{q}{4\pi\epsilon r}$ | $10^{-2}$ | $10^{-1}$ |
| II.6.11 | $V_e = \frac{1}{4\pi\epsilon}\frac{p_d\cos(\theta)}{r^2}$ | $10^{-3}$ | $10^{-1}$ |
| II.6.15a | $E_f = \frac{3}{4\pi\epsilon}\frac{p_d z}{r^5}\sqrt{x^2 + y^2}$ | $10^{-3}$ | $10^{-2}$ |
| II.6.15b | $E_f = \frac{3}{4\pi\epsilon}\frac{p_d}{r^3}\cos\theta\sin\theta$ | $10^{-2}$ | $10^{-2}$ |
| II.8.7 | $E = \frac{3}{5}\frac{q^2}{4\pi\epsilon d}$ | $10^{-2}$ | $10^{-1}$ |
| II.8.31 | $E_{den} = \frac{\epsilon E_f^2}{2}$ | $10^{-2}$ | $10^{-1}$ |
| II.10.9 | $E_f = \frac{\sigma_{den}}{\epsilon}\frac{1}{1+\chi}$ | $10^{-2}$ | $10^{-1}$ |
| II.11.3 | $x = \frac{qE_f}{m(\omega_0^2 - \omega^2)}$ | $10^{-3}$ | $10^{-2}$ |
| II.11.7 | $n = n_0(1 + \frac{p_d E_f \cos\theta}{k_b T})$ | $10^{-2}$ | $10^{-1}$ |
| II.11.20 | $P_* = \frac{n_\rho p_d^2 E_f}{3k_b T}$ | $10^{-3}$ | $10^{-1}$ |
| II.11.27 | $P_* = \frac{n\alpha}{1 - n\alpha/3}\epsilon E_f$ | $10^{-3}$ | $10^{-1}$ |
| II.11.28 | $\theta = 1 + \frac{n\alpha}{1 - (n\alpha/3)}$ | $10^{-4}$ | $10^{-2}$ |
| II.13.17 | $B = \frac{1}{4\pi\epsilon c^2}\frac{2I}{r}$ | $10^{-2}$ | $10^{-1}$ |
| II.13.23 | $\rho_c = \frac{\rho_{c_0}}{\sqrt{1 - v^2/c^2}}$ | $10^{-4}$ | $10^{-2}$ |
| II.13.24 | $j = \frac{\rho_{c_0} v}{\sqrt{1 - v^2/c^2}}$ | $10^{-4}$ | $10^{-1}$ |
| II.15.4 | $E = -\mu_M B\cos\theta$ | $10^{-3}$ | $10^{-1}$ |
| II.15.5 | $E = -p_d E_f \cos\theta$ | $10^{-3}$ | $10^{-1}$ |
| II.21.32 | $V_e = \frac{q}{4\pi\epsilon r(1 - v/c)}$ | $10^{-3}$ | $10^{-1}$ |
| II.24.17 | $k = \sqrt{\frac{\omega^2}{c^2} - \frac{\pi^2}{d^2}}$ | $10^{-5}$ | $10^{-2}$ |
| II.27.16 | $F_E = \epsilon c E_f^2$ | $10^{-2}$ | $10^{-1}$ |
| II.27.18 | $E_{den} = \epsilon E_f^2$ | $10^{-2}$ | $10^{-1}$ |
| II.34.2a | $I = \frac{qv}{2\pi r}$ | $10^{-2}$ | $10^{-1}$ |
| II.34.2 | $\mu_M = \frac{qvr}{2}$ | $10^{-2}$ | $10^{-1}$ |
| II.34.11 | $\omega = \frac{g\, qB}{2m}$ | $10^{-4}$ | $10^{-1}$ |
| II.34.29a | $\mu_M = \frac{qh}{4\pi m}$ | $10^{-2}$ | $10^{-1}$ |
| II.34.29b | $E = \frac{g\,\mu_M B J_z}{\hbar}$ | $10^{-4}$ | $10^{-1}$ |
| II.35.18 | $n = \frac{n_0}{\exp(\mu_m B/(k_b T)) + \exp(-\mu_m B/(k_b T))}$ | $10^{-2}$ | $10^{-2}$ |
| II.35.21 | $M = n_\rho \mu_M \tanh(\frac{\mu_M B}{k_b T})$ | $10^{-4}$ | $10^{-4}$ |
| II.36.38 | $f = \frac{\mu_m B}{k_b T} + \frac{\mu_m \alpha M}{\epsilon c^2 k_b T}$ | $10^{-2}$ | $10^{-1}$ |
| II.37.1 | $E = \mu_M(1+\chi)B$ | $10^{-3}$ | $10^{-1}$ |
| II.38.3 | $F = \frac{YAx}{d}$ | $10^{-3}$ | $10^{-1}$ |
| II.38.14 | $\mu_S = \frac{Y}{2(1+\sigma)}$ | $10^{-3}$ | $10^{-1}$ |
| III.4.32 | $n = \frac{1}{e^{\frac{\hbar\omega}{k_b T}} - 1}$ | $10^{-3}$ | $10^{-2}$ |
| III.4.33 | $E = \frac{\hbar\omega}{e^{\frac{\hbar\omega}{k_b T}} - 1}$ | $10^{-3}$ | $10^{-3}$ |
| III.7.38 | $\omega = \frac{2\mu_M B}{\hbar}$ | $10^{-2}$ | $10^{-1}$ |
| III.8.54 | $p_\gamma = \sin(\frac{Et}{\hbar})^2$ | $10^{-3}$ | $10^{-3}$ |
| III.9.52 | $p_\gamma = \frac{\frac{p_d E_f t}{\hbar}\sin((\omega-\omega_0)t/2)^2}{((\omega-\omega_0)t/2)^2}$ | $10^{-3}$ | $10^{-1}$ |
| III.10.19 | $E = \mu_M\sqrt{B_x^2 + B_y^2 + B_z^2}$ | $10^{-4}$ | $10^{-1}$ |
| III.12.43 | $L = n\hbar$ | $10^{-3}$ | $10^{-1}$ |
| III.13.18 | $v = \frac{2Ed^2 k}{\hbar}$ | $10^{-4}$ | $10^{-1}$ |
| III.14.14 | $I = I_0(e^{\frac{qV_e}{k_b T}} - 1)$ | $10^{-3}$ | $10^{-1}$ |
| III.15.12 | $E = 2U(1 - \cos(kd))$ | $10^{-4}$ | $10^{-1}$ |
| III.15.14 | $m = \frac{\hbar^2}{2Ed^2}$ | $10^{-2}$ | $10^{-1}$ |
| III.15.27 | $k = \frac{2\pi\alpha}{nd}$ | $10^{-3}$ | $10^{-1}$ |
| III.17.37 | $f = \beta(1 + \alpha\cos\theta)$ | $10^{-3}$ | $10^{-1}$ |
| III.19.51 | $E = \frac{-mq^4}{2(4\pi\epsilon)^2\hbar^2}\frac{1}{n^2}$ | $10^{-5}$ | $10^{-2}$ |
| III.21.20 | $j = \frac{-\rho_{c_0} q A_{vec}}{m}$ | $10^{-2}$ | $10^{-1}$ |

Table 8: Tested Equations, part 2.