[Reviews · NeurIPS 2020]

Review 1

Summary and Contributions: The paper describes improvements to AI Feynman, a system for symbolic regression of real-valued functions of (small numbers of) real-valued inputs. The system heuristically detects several forms of modularity in the unknown function's computation graph by examining its gradient, as estimated from a neural net approximation to the function. The search over candidate expressions is pruned by discarding all candidates not on the Pareto frontier of accuracy (measured in error description length) at a given level of complexity. Experiments suggest that these improvements make the system more robust to noise and allow it to solve problems that the previous version could not solve.

Strengths: It seems possible that effective symbolic regression would have widespread scientific utility, although the task has been only sporadically studied in mainstream ML communities. The presented approach of minimizing the total description length of (the formula + errors) is highly principled. Examining the gradients of a fitted surrogate function is an insightful and to my knowledge novel approach to detecting modularity. The empirical results appear to substantially improve on the previous system, which (we are told) was already state of the art. In addition, symbolic regression of probability distributions is an interesting and novel task, and the proposed flow-based approach is natural and seems promising.

Weaknesses: The system itself is seemingly quite complex, full of heuristics and relatively slow: this may be the state of the art, but there's likely quite a lot of room for improvement. The results are positive, but they take essentially the form of a victory lap which gives the paper something of an air of 'salesmanship' and makes it difficult to evaluate them totally objectively. It would be more interesting to see some failure cases and areas for improvement. (perhaps those can be added in a camera-ready after the salesmanship has fulfilled its purpose). Like the previous work, the evaluation is on examples selected by the authors from the physics literature. I'm not suggesting that these are cherrypicked, but any published formulas are likely to already be expressed in the 'right' coordinates for the problem being solved and thus present an easier task than what might be encountered in practice. In future work it would be impressive (and hopefully possible) to illustrate a proof-of-concept workflow in which a system like this one actually discovers a novel and useful result from empirical data. I would also like to see more discussion of tradeoffs around the neural surrogates f_NN. We might hope that symbolic regression would be effective at recovering structure that a neural net *wouldn't* fit, and especially at extrapolating from a relatively small number of observations, but presumably that is not the case with this strategy. The paper briefly mentions that the neural fit can be poor (lines 108-110); as far as I can tell the issue is 'addressed' by just not running any experiments in regimes where that would be expected to be the case. Presumably one needs a fairly high-capacity network to fit arbitrary symbolic functions---how is overfitting addressed? Does the system inherently require thousands of data points in order to get a reasonable fit?

Correctness: I found no obvious correctness flaws.

Clarity: The paper is easy to follow and does an excellent job of expressing concepts clearly and concisely.

Relation to Prior Work: The modifications from the previous AI Feynman work are presented relatively clearly. I had not read the previous paper, though (after skimming it) it seems like relatively important context for the techniques and results presented in this one.

Reproducibility: No

Additional Feedback: I am not a huge fan of the name "AI Feynman": it combines hero-worship of a famous physicist while (somehow at the same time) selling his contributions short: free-form fitting of formulas to data may be a tool in the theoretical-physics toolbox, but it's not what Feynman or most physicists are famous for. Formulas themselves are interesting only insofar as they're built on top of concepts that 'carve the world at the joints'; great physicists discover such concepts, and there's no claim that this work can do anything like that. Given that he seems to be the canonical example of this sort of work, perhaps 'AI Kepler' would be an equally evocative and more fitting name for a future incarnation of this system? (or of course one might also prefer to look for a naming scheme not based on dead white men). ------------------------------ Thanks to the authors for clarifications in the rebuttal. Overall I enjoyed the paper and believe it should be accepted, although there are a couple of points I'm still confused about: - Do the authors expect the system to function reasonably on small training sets (say, tens of points)? If as the authors suggest the failure mode of an overfitting NN is 'failure to discover formulas', it sounds like the answer might be 'no'. It would be good to discuss this more explicitly. - The rebuttal points out that MDL training provably avoids overfitting. Are the neural nets trained using an MDL objective? If so, that should be mentioned (and might be an interesting paper in its own right---presumably one can't provably *find* the MDL global minimizer in most cases); if not, then it seems like NN overfitting is still a potentially significant issue worth discussing.


Review 2

Summary and Contributions: The authors propose an improved method for symbolic regression: first, they use the available data to train a neural network, second, they use gradient information from the neural network to come up with a symbolic formula that approximates the neural network, using its gradient to decompose the regression problem into smaller problems. While the general method is based on a previous paper, the authors suggest new types of decompositions that allow their improved method to scale to more difficult regression problems and deal with more noise in the input data. Additionally, they return a Pareto front of solutions that capture various accuracy / succinctness tradeoffs and describe how through normalising flows their method can also learn from samples of distributions (rather than input-output pairs).

Strengths: - the claims in the paper are sound - symbolic regression is highly relevant and they improve upon the state of the art - the paper is well written and illustrated -- I commend the authors for making their approach so accessible

Weaknesses: W1 Generally, the paper is very interesting, however, it also leans on the ideas of its predecessor. This makes the improvements somewhat incremental (obtaining a Pareto front, adding additional types of graph modularity and using normalising flows to tackle learning from samples).

Correctness: The claims seem correct, so does the methodology.

Clarity: Yes (see strengths)!

Relation to Prior Work: Yes, albeit that the differences to the previous AI Feynman could be a bit better explained.

Reproducibility: No

Additional Feedback: Reproducibility: how is f_{NN} trained? - the formulas in Figure 1 are a bit hard to read - the commas in the subscripts in the Action of Generalized symmetry (Table 1) are confusing Post rebuttal: The authors response was adequate and if they follow through on their promises I think the paper will address many of my key concerns. The other reviews gave interesting perspectives but also confirms my belief that this paper should be accepted. Combined with the excellent quality of the writing and the promise of the authors to follow up on the criticisms raised during the reviewing process I stick to my original score.


Review 3

Summary and Contributions: The paper introduce a novel framework based on Feynman for symbolic regression by exploiting graph molecularity with several robust improvements and speedups.

Strengths: 1. It introduce a mean-error-description-length as a measurement of prediction errors to avoid overfittings. 2. It leverages normalizing flows to estimate an unknown probability distribution.

Weaknesses: 1. When use neural network to approximate unknown function, how to avoid overfiting during the training of neural network? 2. If the neural network perfectly approximate the observed y due to overfitting on noise, will the proposed method be misled by the overfitting NNs?

Correctness: Yes

Clarity: In the caption of Figure 1 in appendix, what is the Figure ??. Also, legend is needed in the figure.

Relation to Prior Work: Not sure.

Reproducibility: Yes

Additional Feedback:

[Author Response · NeurIPS 2020]

We wish to thank all three referees for their encouraging and helpful reports. We think all the suggestions and comments are excellent, and plan to exploit the space available thanks to the additional 9th page to expand and improve the paper to address them as described below.

**Reproducibility and code release (reviewers 1 & 2):** We will significantly expand the description of the method to make it easier to reproduce. To ensure reproducibility, we commit to publicly releasing our code on GitHub and via the Python Package Index PyPI (so that it can be pip-installed) on acceptance.

**Clarification of relation to prior work (reviewers 1, 2 & 3):** We will expand the description of how our method differs from and improves upon prior work (aside from deploying normalizing flows flows to enable regression of probability distributions from samples). Although the original AI Feynman paper used a neural network approximation of the unknown function to discover simplifying function properties, it did so in an unprincipled and ad hoc way that we replace by a general, principled and much more effective method:

1. Instead of discovering merely two types of graph modularity (symmetry and separability) involving merely four particular bivariate functions ($+$, $-$, $\times$ and $\div$), our method has the potential to discover *any* graph modularity involving *any* functions of $n = 2, 3, ...$ variables. A key contribution is showing how this can be efficiently done by examining gradients of the neural network fit.

2. Instead of concluding that a candidate function or graph decomposition is good because the fitting accuracy exceeds arbitrary hyperparameter-determined thresholds, we eliminate these hyperparameters and use a Patero-frontier (of description-length complexity versus accuracy) to prune our search over candidate expressions by discarding all candidates not on the frontier, improving robustness to noise by 1-3 orders of magnitude.

3. Instead of simply rejecting formula candidates using $L_\infty$-norm (rejecting as soon as the error for a single data point crosses a threshold), we reject using statistical hypothesis testing, improving robustness to noise.

**Expanded discussion of NN architecture, training, overfitting (reviewers 1, 2 & 3):**

We will significantly expand the description of our neural network architecture and training, enabled by the expanded page limit. We will also add an extensive discussion of how overfitting impacts our method — here are the highlights:

1. To minimize neural network overfitting, we early-stop training when the validation loss starts increasing.

2. We avoid using our neural network (to guess symbolic functions) outside the domain where it was trained.

3. Overfitting noise would generally *reduce* apparent graph modularity, thus causing failure to discover formulas rather than discovery of spurious "overfit" formulas.

4. A key desirable feature of the minimum-description-length formalism (the information-theoretical inspiration for our method) is that it provably avoids overfitting as shown in refs [28,30].

**Expanded discussion of limitations, failure cases and areas for improvement (reviewer 1):** Text will be added clarifying that there is great room for further improvement, both to symbolic regression itself and to the problem of identifying a low-dimensional latent space from high-dimensional data whose coordinates are promising candidates for symbolic regression. To help readers identify such opportunities for future work, text will be added providing more details on failure modes of our method.

**Algorithm name (reviewer 1):** Clarification: The name "AI Feynman" originates from the benchmark dataset upon which our algorithm was originally trained (100 famous/complex equations from the Feynman Lectures on Physics trilogy).

**Quick fixes (reviewers 2 & 3):**

1. We will expand the labels in Figure 1.

2. We will improve the confusing notation for the action of generalized symmetry (Table 1).

3. We will fix the legend and the "??" in Figure 1 in the appendix.

[Meta-Review · NeurIPS 2020]

This paper presents a nice advance in symbolic regression, it is well-written, and the reviewers were unanimously enthusiastic about it. Assuming the authors revise the paper as promised in the author feedback, this will be a very strong paper for NeurIPS.